# Uptake and Distribution of Administered Bone Marrow Mesenchymal Stem Cell Extracellular Vesicles in Retina

**DOI:** 10.3390/cells10040730

**Published:** 2021-03-25

**Authors:** Biji Mathew, Leianne A. Torres, Lorea Gamboa Acha, Sophie Tran, Alice Liu, Raj Patel, Mohansrinivas Chennakesavalu, Anagha Aneesh, Chun-Chieh Huang, Douglas L. Feinstein, Shafigh Mehraeen, Sriram Ravindran, Steven Roth

**Affiliations:** 1Department of Anesthesiology, College of Medicine, University of Illinois at Chicago, Chicago, IL 60612, USA; bijim@uic.edu (B.M.); ltorre6@uic.edu (L.A.T.); lgamboaa@uic.edu (L.G.A.); stran4@uic.edu (S.T.); aliu50@uic.edu (A.L.); rpate334@uic.edu (R.P.); mchenn2@uchicago.edu (M.C.); aaneesh@haverford.edu (A.A.); dlfeins@uic.edu (D.L.F.); 2Department of Oral Biology, College of Dentistry, University of Illinois at Chicago, Chicago, IL 60612, USA; chuang43@uic.edu (C.-C.H.); sravin1@uic.edu (S.R.); 3Jessie Brown Veterans Administration Medical Center, Department of Anesthesiology, Chicago, IL 60612, USA; 4Department of Chemical Engineering, College of Engineering, University of Illinois at Chicago, Chicago, IL 60607, USA; tranzabi@uic.edu

**Keywords:** astrocytes, exosomes, extracellular vesicles, in vivo imaging, ischemia, microglia, retina, retinal ganglion cells

## Abstract

Cell replacement therapy using mesenchymal (MSC) and other stem cells has been evaluated for diabetic retinopathy and glaucoma. This approach has significant limitations, including few cells integrated, aberrant growth, and surgical complications. Mesenchymal Stem Cell Exosomes/Extracellular Vesicles (MSC EVs), which include exosomes and microvesicles, are an emerging alternative, promoting immunomodulation, repair, and regeneration by mediating MSC’s paracrine effects. For the clinical translation of EV therapy, it is important to determine the cellular destination and time course of EV uptake in the retina following administration. Here, we tested the cellular fate of EVs using in vivo rat retinas, ex vivo retinal explant, and primary retinal cells. Intravitreally administered fluorescent EVs were rapidly cleared from the vitreous. Retinal ganglion cells (RGCs) had maximal EV fluorescence at 14 days post administration, and microglia at 7 days. Both in vivo and in the explant model, most EVs were no deeper than the inner nuclear layer. Retinal astrocytes, microglia, and mixed neurons in vitro endocytosed EVs in a dose-dependent manner. Thus, our results indicate that intravitreal EVs are suited for the treatment of retinal diseases affecting the inner retina. Modification of the EV surface should be considered for maintaining EVs in the vitreous for prolonged delivery.

## 1. Introduction

Cell replacement therapy using mesenchymal (MSC) and other stem cells has been evaluated to treat diabetic retinopathy and glaucoma [1]. While attractive, this approach has significant limitations, including few cells integrated into the retina, aberrant growth, and surgical complications [2]. Recent studies have shown that Mesenchymal Stem Cell Exosomes/Extracellular Vesicles (MSC EVs), which include exosomes and microvesicles, are in fact largely responsible for the paracrine effects of MSCs [3]. As a result, EVs are rapidly emerging as an alternative tool because they promote immunomodulation, repair, and regeneration by mediating the paracrine effects of MSCs [4,5]. The therapeutic efficacy of EVs derived from various precursor cell types hasbeen demonstrated in models of injury in a variety of organs, including myocardium [6], brain [7], kidney [8], and lung [9]. MSC EVs trigger specific cellular responses, with micro RNA (miRNA) from EVs playing a key role [10]. Compared to MSCs, their EVs have the unique advantages of being non-immunogenic and non-tumorigenic, and they are readily endocytosed by target cells to exert specific cellular effects [11]. EVs are a novel, effective means to deliver therapeutic molecules as well as their intrinsic neuroprotective, anti-inflammatory, and vascular-protective properties to the retina [4]. Two recent reviews highlighted the potential for EV therapy in the retina [12,13].

Intravitreal MSC EVs attenuated nerve fiber layer and RGC functional loss in a rat model of glaucoma, and in optic nerve crush injury [14]. We demonstrated that MSC EVs prevented functional loss, and suppressed apoptosis and neuro-inflammation in acute retinal ischemia in rats [11]. Agents injected into the vitreous can directly access retinal cells and achieve higher concentrations vs. systemic administration. Thus, intravitreal injection is now a standard clinical procedure in age-related macular degeneration (AMD) or diabetic retinopathy for the administration of anti-vascular endothelial growth factor (VEGF) therapy [15]. However, the movement of EVs within the vitreous, retina, and cells that take up injected EVs has not been determined.

The vitreous humor is mainly water with structural components, type II collagen, hyaluronic acid, glyco-aminoglycans, heparin sulfate, and chondroitin sulfate [16]. EVs bind to extracellular matrix components, including fibronectin, via proteins, such as α4β1 integrins [16]. This suggests that EVs binding to vitreous components could serve as a reservoir for EV delivery to the retina. To access the retina, particles must cross the retinal inner limiting membrane (ILM) of collagen IV, laminin, and fibronectin connected by proteoglycans [17]. Apaolaza showed that 90 nm gold nanoparticles (NPs) transited the ILM in retinal explants [18]; thus, EVs, with a size range of 100–200 nm, would be expected to cross the ILM into the retina. Accordingly, we hypothesized that EVs would demonstrate prolonged residence in the vitreous and delivery into the retina after intravitreal injection.

Our study objectives included determining the cells taking up EVs as well as the time course of uptake. EV uptake was determined in multiple cell types in the retina following intravitreal administration. We used in vitro and in vivo systems to study EVs in retinal neurons, astrocytes, and microglia, relevant translationally to the treatment of blinding diseases, including glaucoma, AMD, and diabetic retinopathy [11]. These data are essential for ensuring the future clinical translatability of EV therapy. We also sought to determine the vitreous kinetics of injected EVs, relevant to studies aiming to optimize intravitreal injection treatment strategies for EVs. We also investigated the toxicity of injected EVs, a necessary component of moving EV therapy toward clinical translation.

## 2. Materials and Methods

### 2.1. Culture of Human MSCs and Isolation of EVs

We previously described the procedures for preparing EVs from human MSCs [11,19]. In brief, human bone marrow-derived MSCs (hMSCs) from Lonza (PT-2501, Basel, Switzerland) were cultured in α-MEM/20% FBS, 1% L-Glutamine, and 1% antibiotic-anti-mycotic solution (Thermo-Fisher, Waltham, MA, USA). After seeding to confluence, EVs were isolated, as we reported previously [20]. Briefly, cultures were washed with serum-free medium and cultured for 48 h under normoxia (21% O_2_). Conditioned medium was collected, then whole cells and debris were removed by centrifugation. Supernatant was filtered with a 0.22 μm pore filter, transferred to a 100 kDa cut-off ultra-filtration tube (Amicon Ultra-15, Millipore, Burlington, MA, USA), then centrifuged (3000× g) at 4 °C for 45 min. EVs were isolated using Exo Quick-TC EV Precipitation Solution (EXOTC10A-1, System Biosciences, Palo Alto, CA, USA), [21] then resuspended in phosphate-buffered saline (PBS). In earlier studies, it was shown that precipitation-isolated EVs were of a similar size to those prepared by ultracentrifugation, contain the same markers and micro RNA (miRNA), and were functionally active [22,23,24]. Suspensions were normalized to the cell number from the tissue culture plate, and diluted. As we previously reported, the majority of EVs isolated from MSCs were exosomes [11,25]. While the 0.22 μm filter reduced contaminants, EVs isolated from MSCs using this polytheylene glycol-based method may also precipitate some lipoproteins, protein aggregates, and apoptotic bodies [26].

EVs were labeled using green-fluorescent Exo-Glo Protein labeling reagent (EXOGP300A1, System Biosciences) as previously reported [27]. Exo-Glo labels amine groups of internal proteins of EVs using carboxy-fluorescein diacetate succinimidyl ester (CFSE) chemistry [11,21,28]. Unlike lipid and RNA-binding dyes, CFSE does not undergo non-specific aggregation; rather, it stains single EVs, and does not alter the size distribution or concentration of EVs [29]. MSC EVs were incubated with Exo-Glo for 20 min at 37 °C. Labeled MSC EVs were then precipitated by adding Exo Quick-TC, incubated overnight at 4 °C, and centrifuged for 10 min at 10,000 rpm. The pellet was re-suspended in PBS.

### 2.2. In Vivo Administration of EVs

Procedures conformed to the Association for Research in Vision and Ophthalmology Resolution on the Use of Animals in Research and were approved by our Animal Care Committee. Experiments were conducted during daylight hours. Using Sprague Dawley rats (200–250 g; Harlan, Indianapolis, IN, USA) maintained on a 12 h on/12 h off light cycle, we injected Exo-Glo labeled hMSC EVs (4 μL of 1 × 10^9^ particles/mL) into the mid-vitreous, as we previously described [11]. Eyes were removed at 1, 7, 14, and 28 days after injection to determine the time course of uptake by retinal cells.

### 2.3. Immunostaining of Retinal Flat Mounts and Cryosections

For flat mounting, after euthanasia and whole animal perfusion-fixation with PBS and 4% paraformaldehyde (PFA) [30], eye cups were prepared by removing the cornea, lens, and vitreous as we previously reported, [11] post-fixed in 4% PFA for 30 min, washed in PBS, permeabilized with phosphate buffered saline/Tween, 0.3% Triton X-100 (PBST), then blocked overnight in 2% Triton X-100, 10% normal serum and 1 mg/mL bovine serum albumin (BSA) [31]. Primary antibodies (Appendix A) were incubated with eyecups at 4 °C for 48 h, followed by washing and incubation with secondary antibodies (Alexa Fluor 555 and 647, Molecular Probes, Thermo-Fisher) for 48 h at 4 °C. After washing, retinal tissues were dissected from the choroid, placed on a glass slide, and mounted with Pro-Long Diamond Antifade Mounting Solution with DAPI (4′,6-diamidino-2-phenylindole, P36962, Life Technologies, Thermo-Fisher). Slides were imaged with a Zeiss (White Plains, NY, USA) 710 Confocal Microscope, and images were de-convoluted with Zeiss Zen v2.4 software.

Cryosections were fixed in 4% PFA for 2 h, and blocked with 10% donkey serum, 0.5% Triton-X, and 1 mg/mL BSA in PBS. Sections were exposed to primary antibodies overnight, followed by secondary antibodies for 30 min at room temperature. Sections were mounted with Prolong Diamond Antifade Mounting Agent containing DAPI.

### 2.4. EV Localization and Quantitation of Uptake and Co-Localization

To determine the uptake of EVs by retinal cells, and to prove intracellular localization, we used specific protein markers for retinal cell types as previously described [32]. The degree of overlap between channels (green-labeled EVs and cell-specific markers) was quantitated on flat mount images using Fiji and associated plug-ins.

Images were tested with the Co-localization Threshold Plugin (https://imagej.net/Colocalization_Threshold, accessed on 28 February 2021). To minimize the impact of background [33], a region of interest (ROI) was used; at least two different ROIs were tested for consistency in each image. We used the Costes Auto-threshold Method from the Co-loc2 plugin (https://imagej.net/Coloc_2, accessed on 28 February 2021), and the Thresholded Manders’ split colocalization coefficients (0 = no colocalization; 1 = perfect colocalization), for linear regression between channels [34]. The Costes non-biased automated method prevents subjectivity that might otherwise result if the thresholds were to be set manually [35]. When Manders’ was > 0, with linear correlation present between channels, we then tested the same ROIs using the Coloc 2 plugin. (alternative co-localization programs, such as JACOP, https://imagej.net/JaCoP (accessed on 28 February 2021), require testing of the entire image and cannot test ROIs, thereby increasing the influence of background and introducing risk of error) [36]. From Coloc 2, we report threshold regression, Costes Pearson’s R, Li’s ICQ [37], Spearman’s correlation, Thresholded Manders’ M1/M2, and Costes P [38].

To quantify the fluorescence intensity of EVs in retinal cells in flat mounts, an outline was drawn in 3-4 ROIs, and the area and mean fluorescence were measured using Fiji. ROIs were measured from non-stained areas in the same images to account for background. The Corrected Total Cell Fluorescence (CTCF) was calculated as CTCF = integrated density − [Area of ROI × (Average Intensity of Background ROIs)/(Average area of Background ROIs)], modified from McCloy et al., to account for variation in the size of the selected area of background [39]. When specific cells were identifiable, e.g., RGCs or microglia, CTCF was divided by the area of the ROI for the cells, resulting in a CTCF/cell intensity. Where specific cells were not identifiable because of overlap of cellular processes, e.g., Muller cells, we divided CTCF by the measured area of the cells. Results were expressed as mean fluorescence +/− SEM, and compared over time using analysis of variance (ANOVA). Results were considered significant for *p* < 0.05.

### 2.5. Retinal Cell Culture

We cultured astrocytes, microglia, and mixed retinal neurons from retinae of newborn rat pups. Primary enriched cultures of rat retinal microglia and astrocytes were prepared from mixed cultures of retinal glial cells as we described previously [40]. Briefly, aseptically collected retinal tissues were mechanically dissociated by pipetting up and down and centrifuged to collect cells, re-suspended in Dulbecco’s Modified Eagle Medium (DMEM): F12 (1:1)/20% fetal bovine serum (FBS)/1% penicillin/streptomycin, and plated. The culture medium was changed within 24 h, and then twice a week until the astrocytes formed a monolayer. At that time, the culture medium was replaced with PBS without Ca^2+^ and Mg^2+^ (Sigma-Aldrich) and the flasks vigorously shaken to remove non-adherent microglia from the astrocyte bed. Microglial cells were plated onto 96-well plates or on cover slips. Astrocytes obtained with this procedure were then passaged twice for the first time in 75-cm^2^ flasks and for the second time directly in multi-well plates used for the experimental procedures, carried out in 1% fetal calf serum (FCS)-DMEM.

Primary neurons were cultured from neonatal E19-P1 rat pups as previously described [41,42,43]. Briefly, retinal tissues were separated from enucleated eyeballs and incubated in Hank’s Balanced Salt Solution (#14025092, Gibco) containing 10 U/mL papain, 0.2 mg/mL L-cysteine, and 0.4% DNase І for 5–8 min at 37 °C. They were transferred to ovomucoid solution containing 0.0.4% DNAse and 1% ovomucoid (#T9253, Sigma-Aldrich) in DPBS (#194146, Gibco) to fully quench residual papain activity. Using P1000, retinal pieces were gently triturated, forming a unicellular suspension. After centrifugation at 200× *g* for 11 min, cells were resuspended in rapidly growing mycobacteria (RGM) medium containing 250 μg/mL epidermal growth factor (EGF, #10770-910, Prep-Tech), 100 μg /mL fibroblast growth factor (FGF, #,10018B, Prep-Tech), 50 μg/mL brain-derived neurotrophic factor (BDNF, #450-02, Prep-Tech), and 100 μg/mL neurotrophic factor-3 (NT-3, #450-03, Prep-Tech), plated on poly-D-Lysine and laminin pre-coated cover slips and incubated at 37 °C with 8% CO_2_. Cells were fed every 3–4 days by removing 50% media and adding the same amount of RGM and used within 2 weeks.

For immunocytochemistry, cells were seeded onto glass coverslips in 6-well tissue culture plates; then, 24 h later, 50 μL fluorescently labeled MSC EVs were added and incubated for 1 h at 37 °C. Coverslips were washed in PBS three times, fixed in 4% neutral buffered formalin, and immuno-labeled as previously described [11,44]. Slides were imaged using a Zeiss LSM 710 confocal microscope.

### 2.6. Evaluation of EV Uptake in Cultured Retinal Cells

We studied the uptake of EVs in vitro in astrocytes, microglia, and mixed retinal neurons as we previously described [11]. Quantitation was performed in 96-well plates, with 30,000 or 50,000 cells/well. At 24 h post seeding, increasing amounts of MSC EVs were added and incubated for 1 h at 37 °C. Wells were washed 3 times in PBS, fixed using 4% neutral buffered formalin, and fluorescence measured using a BioTek (Winooski, VT, USA) plate reader with the appropriate band pass filter sets. Uptake curves were fitted by non-linear regression with GraphPad Prism (San Diego, CA, USA).

### 2.7. Uptake of EVs into Retinal Cells in Explants

An advantage of this system is that it resembles the in vivo retina to a greater extent than retinal cells in culture, and is not influenced by diffusion of EVs through the vitreous nor by retinal blood flow [45]. Explants were collected from postnatal day 1–4 rat pups as previously described [46]. In brief, enucleated whole eyes were immersed in ice-cold HBSS and the retina collected carefully, and peeled away from retinal pigment epithelium. Pieces of retina were transferred to Millipore trans-well filter inserts and positioned with the RGC side up in 300 μL of media. Explants were incubated with EVs (5 μL), and maintained for 6 or 14 days; then, cryosections were prepared for immunostaining.

### 2.8. In Vivo Imaging of EVs in the Vitreous

Rats were injected intraperitoneally with ketamine (35 mg/kg), and xylazine (5 mg/kg), and pupils dilated with 0.5% tropicamide (Alcon), and cyclomydril. Fluorescent fundus images were obtained using a Micron IV Retinal Imaging Microscope (Phoenix Research Labs, Pleasanton, CA, USA) after intravitreal injections as we previously reported [11,47].

### 2.9. Quantitation of In Vivo Release Kinetics of EVs in the Vitreous

We determined fluorescence intensity from in vivo images using Fiji (https://imagej.net/Image_Intensity_Processing, accessed on 28 February 2021). We assumed intensity was proportional to the concentration of EVs diffusing through the vitreous. We hypothesized that the excitation process was short, and thus the rate of production of fluorescence followed the profile for the ordinary level of illumination; that is, $\frac{dC}{dt}=\;\mathrm{rate}\;\mathrm{of}\;\mathrm{production}\;\mathrm{of}\;C\;=\;E\left(t\right)$ (Equation (1)), where $E\left(t\right)$ is the net rate of excitation. We also assumed that the rate of disappearance was proportional to C with a proportionality constant $k_{off}$, which was assumed to be identical to the rate constant associated with the unbinding of EVs from the vitreous. Incorporating the unbinding rate, we defined the net rate of production: $\frac{dC}{dt}=\;E\left(t\right)-\;k_{off}C$. (Equation (2)). To simplify analysis of unbinding kinetics, we further assumed that the initial excitation was proportional to an initial condition at t=0. Thus, from the above, we obtained the following: $C=a\exp\left(-\;k_{off}t\right)+b$ (Equation (3)), where a and b are two constants to be found by fitting, and $bk_{off}$ constitutes the initial excitation. Reciprocal of the rate-constant, $k_{off}$, is often called the natural lifetime, which is the lifetime for the EVs residence in the vitreous, τ. Hence, $\tau =\frac{1}{k_{off}}$ (Equation (4)).

### 2.10. Histology

To examine the impact of EVs on retinal structure, and in particular for possible toxicity of the injected EVs, eyes were enucleated 7 days after intravitreal injection of EVs or control PBS, and placed in Davidson’s fixative (11% glacial acetic acid, 2% neutral buffered formalin, and 32% ethanol in H_2_O) for 24 h, transferred to 70% ethanol for 24 h, and stored in PBS at 4 °C. Eyes were embedded in paraffin, sectioned to 5 μm, and stained (hematoxylin and eosin, H&E); they were then examined by light microscopy and cell layer thickness quantitated similarly to our previous descriptions [47,48]. A standardized region of retina centered 1280 μm from the thinning of the neurofilaments arising from the optic nerve head was used to ensure consistency of counts in comparable retinal eccentricities. Cell layer thicknesses were determined using Micron 2.0 (Westover Scientific, Mill Creek, WA, USA) in 3–5 regions about 100 μm apart on both sides of the optic nerve, in a blinded manner, and averaged.

## 3. Results

### 3.1. Uptake of EVs by Retinal Cells In Vivo after Intravitreal Injection

In retinal flat mounts (Figure 1, Figure 2 and Figure 3), we examined colocalization of the intravitreally administered fluorescent EVs in RGCs with anti-brain-specific homeobox/POU domain protein 3A (Brn3a) to stain nuclei [49], and anti-β-tubulinIII (BT3) for cytoplasm. Microglial cells were stained with anti-ionized calcium-binding adaptor molecule (IBA-1) [50], and astrocytes and Muller cells with anti-vimentin [51]. The staining pattern for EVs was different in the three cell types. EVs stained soma and axons/dendrites of RGCs in a vesicle-appearing staining pattern. EVs were present intracellularly in RGCs, as shown by co-localization with BT3 [52] in Brn3a-co-staining cells. Although visible in RGCs and their projections within a day after injection, staining for EVs in RGCs peaked at 14 d after administration; no EVs were visible at 28 d (Figure 1). RGCs were also stained in flat mounts with anti-RNA-binding protein (RBPMS) [53]. EVs were visible in a vesicle-appearing pattern around or overlapping the RBPMS-staining nucleus as with Brn3a (Figure 4D). The indices of co-localization with BT3 in Brn3a-positive cells (Appendix A) confirmed EV uptake into RGCs.

In microglia (Figure 2), EVs were present within a day of injection, staining intensity peaked at 7 d, and only a few scattered EVs were present at 14 d, with none at 28 d. These EVs were intracellular and in a high concentration, which appeared to completely fill the cytoplasm, [54] and co-localization indices confirmed the intracellular uptake of EVs (Appendix A). On cryosections, microglia-containing EVs were confined to the RGC and inner plexiform layers. (Figure 4B). EVs co-localized in a small number of astrocytes and Muller cells (Figure 3) stained with vimentin, and there was no change in the degree of staining for EVs over 28 days. Co-localization indices confirmed the intracellular uptake of EVs into astrocytes (Appendix A). EVs were concentrated in the foot plate regions of the Muller cells, confirmed by also staining with anti-glial fibrillary acidic protein (GFAP) in cryosections (Figure 4B) [55]. Most injected EVs were in the inner plexiform, and inner nuclear layers, with very few in the outer retinal layers (Figure 4A–C). Injected EVs did not significantly alter the thickness of the retinal cell layers (Figure 5, Table 1).

### 3.2. EV Fluorescence Kinetics in the Vitreous after Injection

EV fluorescence in vitreous peaked one day after injection. The unbinding kinetics of EVs closely followed the model proposed in Equation (3). To find the EV fluorescence lifetime, we fit Equation (3) to the fluorescence intensity obtained from in vivo images (Figure 6). Likewise, in our model, the constant b in Equation (3) accounted for the concentration of EVs that remained in the vitreous for >14 days. This model suggests that the lifetime of EVs in vitreous, which is identical to its unbinding time constant, is 2.5 days.

### 3.3. Uptake of EVs into Retinal Cells In Vitro

Astrocytes, microglia, and retinal neurons demonstrated dose-dependent saturable EV uptake kinetics. (Figure 7). Microglia showed dense cytoplasmic staining for EVs, concentrated in large vesicles (Figure 8A). Astrocytes showed smaller appearing collections of EVs (Figure 8B). In mixed retinal neurons, EVs were mainly taken up by RGCs (Figure 9) (also see Appendix A).

### 3.4. EVs in Retinal Explants

Cryosections at 6 and 14 days showed EVs in RGCs, astrocytes, Muller cells, and microglia (Figure 10), similar to our findings in the in vivo retina. As expected, since the neural retinae of explants degenerate, at 14 days, the retinae appeared thinned vs. those at 6 days [56]. Few EVs were located deeper than the inner nuclear layer.

## 4. Discussion

A number of investigators have reported that in several different organ systems, EVs derived from MSCs appear to be a safe alternative to the MSCs with a lower risk of oncogenic transformation and immune reactions, and they can be altered to incorporate therapeutic agents [4]. With the ability to carry therapeutic molecules directly into cells, EVs are a potential alternative to viral gene therapy, the latter recently shown to stimulate the immune system, thereby limiting its optimal dosing and efficacy [57]. EV therapy in the retina has been reported in a small number of studies, including mouse and rat glaucoma, oxygen-induced retinopathy, and in acute ischemic injury in the retina, most showing promising improvements in the function or maintenance of retinal neurons under these stress conditions [11,14,58,59]. Our previous study found that EVs readily entered retinal neurons via an HSPG- and caveolin-dependent mechanism [11]. Prior studies have not examined the time course or uptake of EVs into retinal cells. The present study provides an understanding of the cellular mechanisms of actions of EVs in retina, and necessary background data toward the therapeutic use of EVs in retinal diseases.

Most studies of EVs for therapeutics involve systemic administration, which is limited by the proportion of blood supply to the tissue; barriers to diffusion, particularly the endothelium; and uptake and destruction by the immune system [60]. Direct administration into the vitreous, therefore, offers significant advantages for the treatment of retinal diseases. To our knowledge, there have not been any previous studies of cellular uptake and vitreous kinetics of EVs in the eye after intravitreal injection. Negatively charged phosphatidylserine, which may be associated with different intracellular sites of origin and function [61], is present on some classes of EVs, including those derived from MSCs. The negative charge is expected to enhance mobility through the vitreous. This has not been directly tested, but the movement of nanoparticles (NPs) through the vitreous is influenced by their size relative to the vitreous pore size; thus, synthetic NPs < 500 nm in size diffused more rapidly vs. particles >1100 nm, fitting an estimated average vitreous mesh pore size of 550 nm [62]. Positively charged NPs showed limited mobility in the vitreous. A large portion of positively charged NPs were immobilized and trapped within the injection site, while anionic nanoparticles demonstrated much greater mobility, due to electrostatic interactions with negatively charged hyaluronic acid. In contrast, polyethylene glycol (PEG)ylated nanoparticles, with no electrostatic charge, diffused more rapidly [63]. These principles of nanoparticle movement in the vitreous have also been predicted by computer simulation [64]. Accordingly, from our results, the EVs appeared to readily migrate through the vitreous and into the retina. Despite the capacity of EVs to bind vitreous proteins including fibronectin, most EVs were cleared out of the vitreous within a week after administration.

Three similar studies to ours in rodent brain tissue found that within 2 and 6 h after intranasal administration, EVs were widely distributed into the somatosensory and pre-frontal cortex, amygdala, dentate gyrus, and cerebellum [65,66]. Similar to our findings, EVs were in neurons, microglia, and astrocytes [66], and completely cleared from brain within 24 h [67].

We found that in rat retinas, following the injection of labeled EVs into the vitreous, they were taken up by RGCs, astrocytes, and microglia. We confirmed the findings in a retinal explant model. The staining intensity in the cells suggested that RGCs had higher relative fluorescence than astrocytes and microglia, and also retained EVs for a longer period. However, few EVs were found deeper than in the inner nuclear layer in vivo or in the ex vivo explant. An important proviso, however, when comparing in vivo results to those in the explant and in cell culture, is that in the latter two, the retina was removed from its environment, including the optic nerve, and in essence, the explant and cell culture models already had a degree of retinal injury vs. the normal retinal environment of the in vivo model. In future studies for further translational relevance, it will be important to examine the uptake of EVs into the injured retina in vivo.

EVs readily enter cells, and as we show here, in a dose-dependent and saturable manner, and it is likely that most uptake occurs quickly into the first cells they encounter after crossing the inner limiting membrane from the vitreous. This could explain their uptake into superficially located cells including RGCs, nerve fiber layer, Muller cell endplates, and microglia, leaving few EVs to penetrate deeper into the retina. The EVs showed no retinal toxicity, and the present results are commensurate with our previous demonstration of no effect of EVs on retinal function in vivo, and no immune reaction in rats to EVs from human MSCs [11]. The results are translationally relevant, as EVs may easily access retinal cells that are affected by an array of diseases, including glaucoma, diabetic retinopathy, and AMD.

In vivo, the uptake of EVs is affected by considerably more factors than in vitro, depending upon the concentration of EVs, diffusion distance, cellular density, and pH of the microenvironment and the cells encountered by EVs as they first move through the tissue [68,69]. The uptake of EVs into cells in vitro may not translate into the same in vivo behavior. To date, the mechanisms of endocytosis of EVs in microglia and astrocytes have not been determined, although we did show previously that retinal neurons took up EVs via HSPGs and caveolins [11]. Here, we show that astrocytes, microglia, and retinal neurons take up EVs via dose-dependent saturable kinetics. Accumulating data suggest that EVs alter the pro-inflammatory properties of microglia [70], including a shift into M2-anti-inflammatory conditions [71]. Additionally, it has been shown recently that microglia and astrocytes communicate in part via the release of EVs from microglia [72]. It is an intriguing possibility that, in vivo, microglia or astrocytes released the administered EVs for later uptake into RGCs, although in our present study, we cannot make any conclusions on this possible exchange mechanism.

In the retinal explant, there was expected cellular layer thinning due to cell loss over time, as well as significant glial reaction. These changes occurred, despite the presence of EVs, which are neuroprotective, as we showed previously. However, these results should be considered in light of the limitation of only a single dose of EVs having been administered. The use of this preparation for the study of the neuroprotective capacity of EVs will likely require higher doses of EVs or repeated administration over time.

We previously showed that EVs bind isolated vitreous humor in a dose-dependent manner [11]. This enables the vitreous to serve as a reservoir for the release of EVs into the retina. However, the decay of fluorescence indicates that most EVs are rapidly cleared from the vitreous. An important implication of these findings is that for therapeutic application, repeated injections will be required or that the surface of EVs will have to be altered to enhance vitreous retention. Since the vitreous consists largely of collagen, it may be feasible to prolong the residence time of EVs by modifying the EV surface with peptide sequences that bind collagen and other extracellular matrix proteins [73].

## 5. Conclusions

In conclusion, EVs injected into the vitreous, or administered in an ex vivo retinal explant, were found in astrocytes, microglia, Muller cells, and retinal neurons. The time course of retention varied in the different retinal cells, with the longest residence time being 14 days in the retinal ganglion cells in vivo. There was no toxicity of the EVs in the retina. These results provide important data for the design of pre-clinical studies of the therapeutic efficacy of intravitreally injected EVs.

## Figures and Tables

**Figure 1 cells-10-00730-f001:**
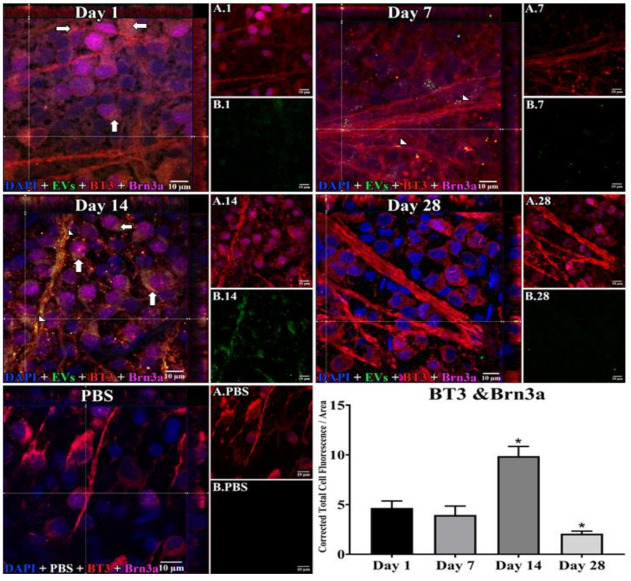
Staining for Extracellular Vesicles (EVs) in retinal ganglion cells (RGCs) in flat mounts of rat retinas. The results show that EV staining in RGCs peaks at 14 days after intravitreal injection of EVs. Fluorescent colors are as follows: anti-β-tubulinIII (BT3, red), anti-Brn3a (magenta), EVs (green), DAPI (blue). Orthogonal 3D projection shown as a composite for each day’s panel, and red and magenta channels (A.1, A.7, A.14, A.28) and green channels (B.1, B.7, B.14, B.28) for a representative z-stack alongside each day’s composite. PBS injected control is bottom left; A.PBS is red and magenta channels, and B.PBS is green channel. Scale bars (10 μm) are on the bottom right of each panel. Arrows = overlap (co-localization, yellow or yellow-orange) of green EVs and anti-β-tubulinIII in Brn3a-containing RGCs. Arrowheads = co-localization of BT3 and EVs in neural projections from RGCs. 100x. Bottom right: graph of green fluorescence intensity (Y axis), expressed relative to background. N = 4 per time point * *p* < 0.05 vs. day 1.

**Figure 2 cells-10-00730-f002:**
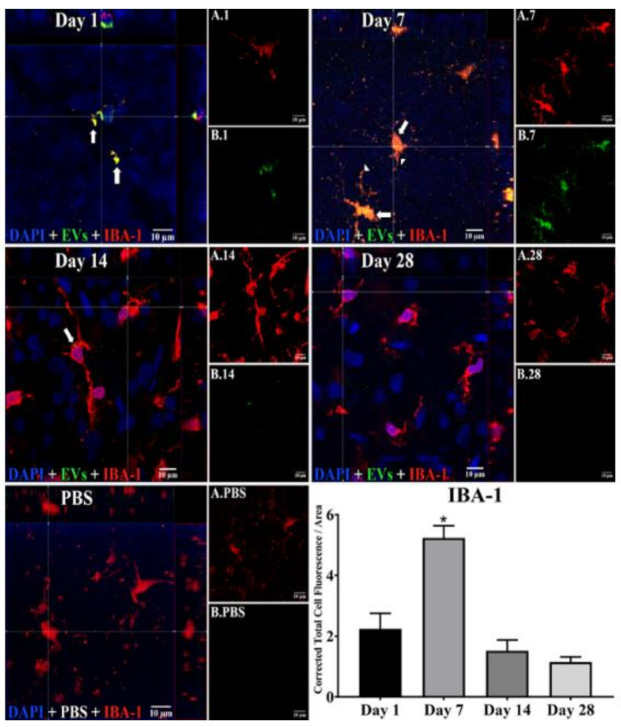
Staining for EVs in microglia in flat mounts of rat retinas. The results show that EV staining in microglia peaks at 7 days after intravitreal injection of EVs. Fluorescent colors are as follows: anti-IBA–1 (red), EVs (green), DAPI (blue). Orthogonal 3D projection shown as a composite for each day’s panel, and red channels (A.1, A.7, A.14, A.28) and green channels (B.1, B.7, B.14, B.28) for a representative z-stack alongside each day’s composite. PBS injected control is bottom left; A.PBS is red channel, and B.PBS is green channel. Scale bars (10 μm) are on the bottom right of each panel. Arrows = overlap (co-localization, yellow or yellow-orange) of green EVs and anti-IBA-1 in microglia. Arrowheads point to vesicular-appearing staining pattern of EVs in microglia. 100x. Bottom right: green fluorescence intensity (*Y* axis) was expressed relative to background. N = 4 per time point * *p* < 0.05 vs. day 1.

**Figure 3 cells-10-00730-f003:**
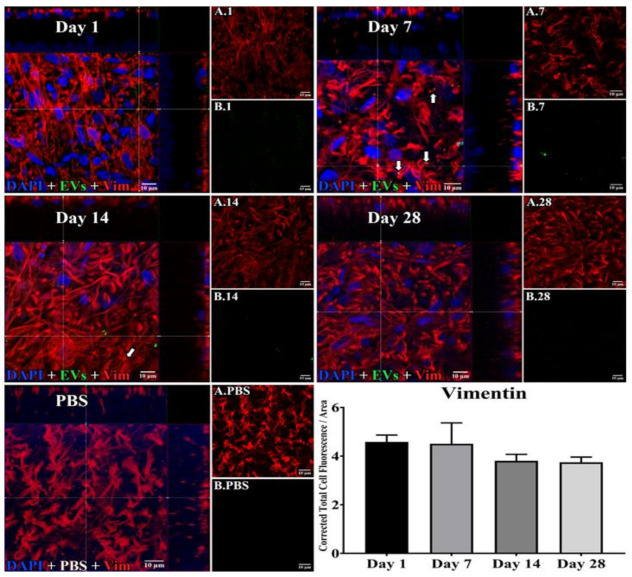
Staining for EVs in astrocytes in flat mounts of rat retinas. The results show that EV staining in astrocytes remained constant after intravitreal injection of EVs. Fluorescent colors are as follows: anti-vimentin (red), EVs (green), DAPI (blue). Orthogonal 3D projection shown as a composite for each day’s panel, and red channels (A.1, A.7, A.14, A.28) and green channels (B.1, B.7, B.14, B.28) for a representative z-stack alongside each day’s composite. PBS injected control is bottom left; A.PBS is red channel, and B.PBS is green channel. Scale bars (10 μm) are on the bottom right of each panel. Arrows = overlap (co-localization, yellow or yellow-orange) of green EVs and anti-vimentin in astrocytes. 100x. Bottom right: green fluorescence intensity (Y axis) was expressed relative to background. N = 4 per time point.

**Figure 4 cells-10-00730-f004:**
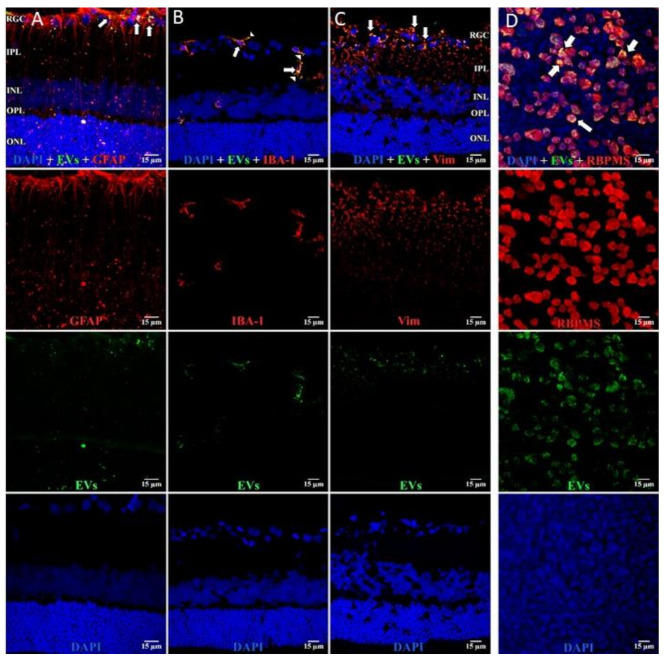
EVs in microglia and in astrocytes and additional confirmation of EVs in RGCs. Retinal 10 m-thick cryosections were prepared 7 days after injection of fluorescent green labeled EVs, and examined using confocal microscopy. Immunostaining for, left to right, (**A**) astrocytes and Muller cell footplates (anti-GFAP), (**B**) microglia (anti-IBA-1), (**C**) cytoplasmic processes of astrocytes and Muller cells (anti-vimentin, “Vim”). DAPI stained nuclei blue. Arrows indicate overlap (yellow color) of green labeled EVs and red (GFAP, IBA-1, or Vim). Arrowhead in (**B**) shows vesicular-appearing EV staining inside microglial cells. Cell layer names provided in (**A**) and (**C**) also apply to (**B**); 40×. RGC = retinal ganglion cell layer; IPL = inner plexiform layer; INL = inner nuclear layer; OPL = outer plexiform layer; ONL = outer nuclear layer. Staining for GFAP and EVs (A) confirms that most EVs in Muller cells are in the footplates. Most staining for EVs in microglia (yellow) in (B), and astrocytes/Muller cells (C), was in the retinal ganglion cell layer and inner plexiform layer. (**D**) Flat mounts with anti-RBPMS (red), EVs (green), and DAPI blue. Nearly all red staining RGCs also contain green EVs (yellow overlap, arrows in top panel in D), providing further support that EVs were taken up by RGCs. Scale bars (15 μm) are on the bottom right of each panel 63×.

**Figure 5 cells-10-00730-f005:**
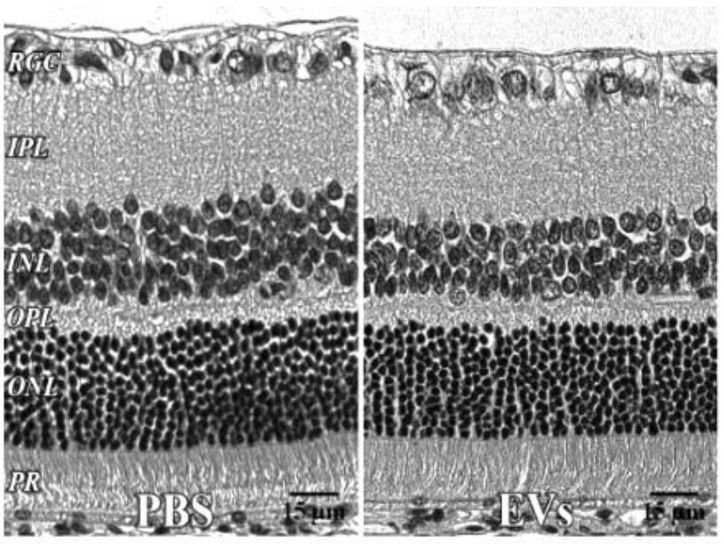
Retinal histology after intravitreal EVs. PBS—injected eye left; EV—injected right, both 7 days after injection, with no significant change in retinal cell layer thicknesses. 40x. RGC = retinal ganglion cell layer; INL = inner nuclear layer; IPL = inner plexiform layer; OPL = outer plexiform layer; ONL = outer nuclear layer; PR = photoreceptor layer. Scale bars (with black lettering, 15 μm) are on bottom right of each panel.

**Figure 6 cells-10-00730-f006:**
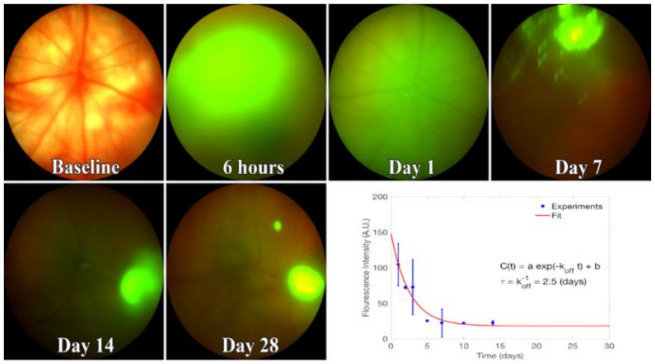
Kinetics of fluorescent EVs in the vitreous. Representative images from 3-4 rats injected with Exo-glo and imaged using Phoenix Micron IV. Bottom right, fluorescence decay curve with calculated parameters. From fitting the curve, the lifetime of EVs in the vitreous is about 2.5 days. Image intensity on Y-axis is mean +/− SD.

**Figure 7 cells-10-00730-f007:**
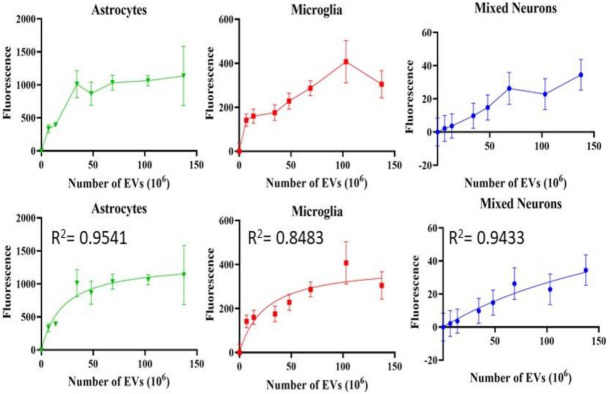
Uptake of EVs by retinal cells in vitro. Fluorescence, Y-axis, is mean +/− SD. X-axis is number of EVs. Bottom row shows the fitted curves using non-linear regression. All demonstrate saturable, dose-dependent endocytosis.

**Figure 8 cells-10-00730-f008:**
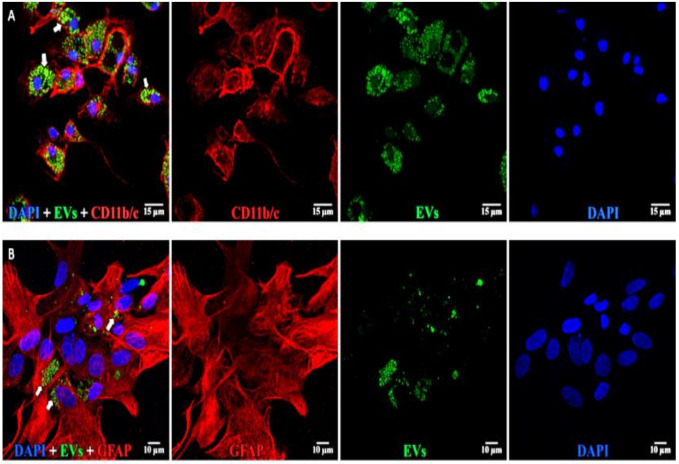
EVs in microglia and astrocytes. (**A**, top) Microglia: left to right: composite, anti-CD11b/c (red), EVs (green), and DAPI (blue). EVs are concentrated in large vesicles (arrows) in the cytoplasm. (**B**, bottom) Astrocytes: left to right: composite, anti-GFAP (red), EVs (green), and DAPI (blue). EVs are small vesicles (arrows) in the cytoplasm. Scale bars (10 μm) are on the bottom right of each panel 63x.

**Figure 9 cells-10-00730-f009:**
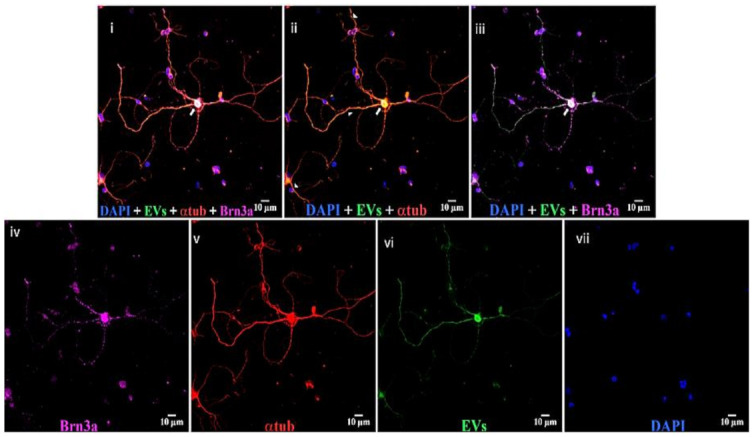
EVs in mixed retinal neurons. Top panels, left to right, (**i**) DAPI (blue), EVs (green), anti-α-tubulin (red), and anti-Brn3a (magenta); (**ii**) DAPI, EVs, and anti-α-tubulin; and (**iii**) DAPI, EVs, and anti-Brn3a. Bottom, (**iv**) anti-Brn3a, (**v**) anti-α-tubulin, (**vi**) EVs, and (**vii**) DAPI. Arrows = overlap (yellow) of α-tubulin and EVs in cytoplasm of RGCs, and arrowheads = yellow overlap in nerve projections from RGCs. White in RGC nucleus in panel iii is overlap of Brn3a magenta and green EVs. Scale bars (10 μm) are on the bottom right of each panel 63x.

**Figure 10 cells-10-00730-f010:**
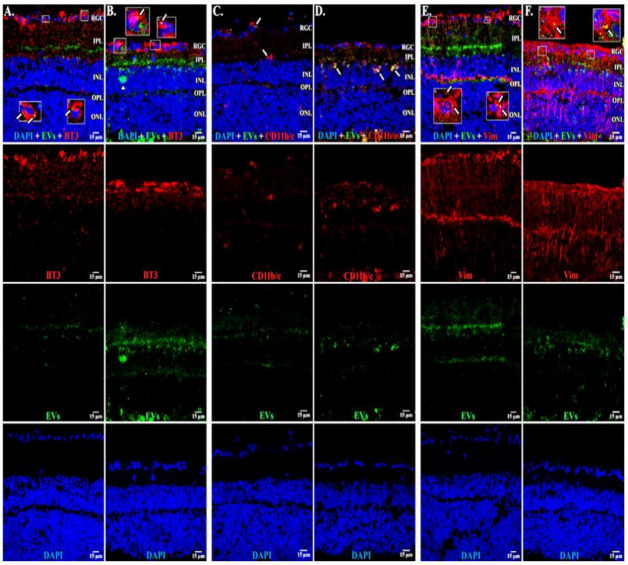
EVs in retinal explants. (**A**,**B**) BT3 (red), EVs (green), and DAPI (blue) at 6 and 14 days after start of incubation with EVs. (**C**,**D**) CD11b/c (red), EVs (green), and DAPI (blue) at 6 and 14 days. (**E**,**F**) vimentin (Vim, red), EVs (green), and DAPI (blue) at 6 and 14 days. Arrows = yellow overlap with EVs; inserts show overlap in detail. RGC = retinal ganglion cell layer; IPL = inner plexiform layer; INL = inner nuclear layer; OPL = outer plexiform layer; ONL = outer nuclear layer. Scale bars (15 μm) are on bottom right of each panel 63x.

**Table 1 cells-10-00730-t001:** Retinal histology after administration of EVs. For the EV toxicity study, histological sections (Figure 5) were stained with hematoxylin and eosin (H and E). One eye received PBS, the other EVs. N = 5. There were no significant changes in retinal layer thicknesses, measured in μm. IPL = inner plexiform layer, OPL = outer plexiform layer, SEM = standard error of the mean.

		PBS Eye		EV Eye	
		Mean	SEM	Mean	SEM
Layer	IPL	44.9	2.6	42.9	0.5
	Inner Layer	102.7	5.6	101.1	0.7
	OPL	9.9	0.4	9.5	0.7
	Outer Layer	82.5	2.3	77.2	4.9
	Total	185.2	7.6	164.1	12.2

## Data Availability

Data are available from the authors upon request.

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
