# Peer review of "Uptake and Distribution of Administered Bone Marrow Mesenchymal Stem Cell Extracellular Vesicles in Retina"

_cells, 2021, doi:10.3390/cells10040730_

Round 1

Reviewer 1 Report

The present paper evaluates the fate of EVs after they are administered into the retina. EVs are a strong candidate therapy for a variety of diseases, including retinal disease, yet how they interact with retina tissue in vivo is largely unknown. The paper thus addresses a very important question and is highly relevant to the field and topical. The data is presented in an easy to read manner and the results are substantial enough to warrant publication. My comments are therefore only on some of the presentation and discussion and are as follows:

The introduction jumps from "cell replacement" to "EVs" without ever highlighting that MSCs have been known to mediate the majority of their effects through paracrine mediated mechanisms. My feeling is that there should be 1-2 statements on the fact that MSCs predominant effects were paracrine mediated, not cell replacement (include papers investigating their secretion of PDGF, NGF, BDNF, NT-3). This then leads on nicely to the focus on EVs.

The authors correctly refer to their isolate as EVs, not exosomes, yet the term exosomes is used throughout the manuscript when referencing other studies (who themselves likely used "exosome"). I think this is enough justification for the authors to include a few sentences on the differences between these vesicles. Since PEG isolates exosomes as well as larger vesicles, this should be highlighted. The precipitation may also include lipoprotein particles, viral particles and protein aggregates. Without a 220um filter, apoptotic bodies are also likely. 

The above comments, as well as more generic aspects of the EV-EYE field are detailed in two recent reviews in PRER (Klingeborn et al 2017 and Mead et al 2020) and should be referenced given the brief introduction.

The authors use several models but never highlight a critical distinction between them. When administered in vivo, the eyes are healthy and uninjured. In the retinal wholemount and retinal cultures, the optic nerve is severed, a necessity to remove the retina. Thus the authors are actually testing EV uptake in 2 scenarios, healthy and injured retina. Ideally, the authors would also test uptake in vivo after optic nerve crush, but at the very least, they should discuss these differences. Importantly, EVs are not going to be delivered into a healthy retina, but an injured one.

Minor points:

Reference 10 is a repeat of reference 53 

Figure 1,2, 3, bottom left, the annotation implied that there was a staining for PBS. I understand that this is a PBS control, but PBS should not be mentioned alongside DAPI and Vimentin etc.

Figure 4,5, 10 . "RGC" should be changed to GCL, to match the other labels (IPL, INL etc..

Author Response

<Thank you for your helpful comments. Our answers appear after each of your points, below. In the manuscript, revisions in response to your comments appear in YELLOW highlighting. 

The introduction jumps from "cell replacement" to "EVs" without ever highlighting that MSCs have been known to mediate the majority of their effects through paracrine mediated mechanisms. My feeling is that there should be 1-2 statements on the fact that MSCs predominant effects were paracrine mediated, not cell replacement (include papers investigating their secretion of PDGF, NGF, BDNF, NT-3). This then leads on nicely to the focus on EVs.

< We have inserted these sentences to enable the Intro to flow right into the focus on EVs. 

The authors correctly refer to their isolate as EVs, not exosomes, yet the term exosomes is used throughout the manuscript when referencing other studies (who themselves likely used "exosome"). I think this is enough justification for the authors to include a few sentences on the differences between these vesicles. Since PEG isolates exosomes as well as larger vesicles, this should be highlighted. The precipitation may also include lipoprotein particles, viral particles and protein aggregates. Without a 220um filter, apoptotic bodies are also likely. 

<We inserted a few sentences concerning this point. 

The above comments, as well as more generic aspects of the EV-EYE field are detailed in two recent reviews in PRER (Klingeborn et al 2017 and Mead et al 2020) and should be referenced given the brief introduction.

<Both references have been added

The authors use several models but never highlight a critical distinction between them. When administered in vivo, the eyes are healthy and uninjured. In the retinal wholemount and retinal cultures, the optic nerve is severed, a necessity to remove the retina. Thus the authors are actually testing EV uptake in 2 scenarios, healthy and injured retina. Ideally, the authors would also test uptake in vivo after optic nerve crush, but at the very least, they should discuss these differences. Importantly, EVs are not going to be delivered into a healthy retina, but an injured one.

<These limitations are now included in the Discussion. 

Minor points:

Reference 10 is a repeat of reference 53 

<Corrected

Figure 1,2, 3, bottom left, the annotation implied that there was a staining for PBS. I understand that this is a PBS control, but PBS should not be mentioned alongside DAPI and Vimentin etc.

<Corrected

Figure 4,5, 10 . "RGC" should be changed to GCL, to match the other labels (IPL, INL etc..

<The "RGC" notation has been spelled out in the figure legend for clarity, to include the word "layer."

Reviewer 2 Report

The purpose of this study was to investigate the uptake of fluorescently labeled extracellular vesicles (EVs) derived from bone marrow-derived mesenchymal stem cells (BM-MSCs) by retinal cells in vivo and in vitro (mixed retinal cell cultures and retinal explants). EVs have the advantage that they do not elicit immunological responses when derived from a different species and can be administered repeatedly. The data show maximum fluorescence in retinal ganglion cells at 14 d, and maximum uptake by microglia at 7 d post intravitreal injection.  A single application of EVs to retinal explants showed uptake, but did not prevent retinal thinning.

General comments:

The is a well-written paper. However, the presentation of the data requires some improvement as the figures presented appear to be too small.

Specific comments:

Page 3, Section 2.4, line 123: “To determine the retinal cells taking up EVs” – proposal: “To determine the uptake of EVs by retinal cells”

Table 1: should be a table on its own and not part of figure 5 (too small numbers).

There are references to supplementary table 1 and 2 withing the manuscript, but the table are no provided for review.

Figures:

Figures (especially Figs. 1-3) appear to be too small in the pdf file to evaluate details. They should at least have page width.

Fig. 3: Vimentin is a very generic marker for astrocytes and Müller cells. A better marker would be GFAP for astrocytes, and CRALBP or glutamine synthetase for Müller cells.

Fig. 4 uses GFAP as a marker for Müller cells, and vimentin as marker of astrocytes. This should be reverse. In normal, non-injured retina, GFAP is mainly a marker for astrocytes and for Müller cells endfeet; and vimentin labels cytoplasmic processes of both. GFAP does not label Müller cell bodies and processes within the retina unless the retina is injured.

Figure 5: the scale bar is difficult to see. Better to make it white, or white with a black border.

Figure 6: the graph is too small (cannot see any labels).

Figure 7: should be larger to see all the labels.

Supplemental Figures 1-3 are not provided for review.

Figure 8 and 9 are somewhat acceptable in size, but Figure 10 is too small to see details.

Author Response

Thank you for your helpful comments. The responses appear below. In the manuscript, except for the figures, any revisions in response to your comments are now indicated using GREEN highlighting. 

General comments:

The is a well-written paper. However, the presentation of the data requires some improvement as the figures presented appear to be too small.

<The figures have been made as large as possible to be consistent with the style requirements of the journal. 

Page 3, Section 2.4, line 123: “To determine the retinal cells taking up EVs” – proposal: “To determine the uptake of EVs by retinal cells”

<The sentence has been re-worded. 

Table 1: should be a table on its own and not part of figure 5 (too small numbers).

<Table 1 has been made larger and is now easily readable

There are references to supplementary table 1 and 2 withing the manuscript, but the table are no provided for review.

<The supplemental tables are provided. 

Figures (especially Figs. 1-3) appear to be too small in the pdf file to evaluate details. They should at least have page width.

Fig. 3: Vimentin is a very generic marker for astrocytes and Müller cells. A better marker would be GFAP for astrocytes, and CRALBP or glutamine synthetase for Müller cells.

Fig. 4 uses GFAP as a marker for Müller cells, and vimentin as marker of astrocytes. This should be reverse. In normal, non-injured retina, GFAP is mainly a marker for astrocytes and for Müller cells end feet; and vimentin labels cytoplasmic processes of both. GFAP does not label Müller cell bodies and processes within the retina unless the retina is injured.

Figure 5: the scale bar is difficult to see. Better to make it white, or white with a black border.

Figure 6: the graph is too small (cannot see any labels).

Figure 7: should be larger to see all the labels.

Supplemental Figures 1-3 are not provided for review.

Figure 8 and 9 are somewhat acceptable in size, but Figure 10 is too small to see details.

<All of the figures and contents therein have been enlarged to the size consistent with the journal's style. These appear to us to be more readable. Points about Muller cell vs astrocyte staining has been revised. 

Reviewer 3 Report

The manuscript describes the uptake and distribution of bone marrow Mesenchymal stem cells Extracellular Vesicles in retinal neurons, Muller cells, astrocytes and microglia. The subject of the paper is adequate for publication in Cells Journal. The overall scientific design of this work is appreciable but the results of the immunofluorescence should be exposed and presented more clearly.

This reviewer will agree with the publication of this manuscript only after a major revision. There are 3 major points and several minor points to be solved, as follows:

  1. The objectives of this study were not clearly stated. It was the intention of the authors to compare into this paper some results of their current work with other previously published results?
  2. The confocal microscopy figures are difficult to understand because the dimensions are too small and do not allow a clear interpretation (Figures 1,2,3,4, 9,10). Furthermore in Figure 6 the fluorescence decay curve is not visible. The captions are very lacking especially in Figures 1 and 2. The control in PBS is mentioned only in Figure 3 and not in Figures 1 and 2. This reviewer recommends showing only the most significant images of each experiment in the article.
  3. In the discussion is not clearly stated whether EVs from hBMSCs exhibit phosphatidylserine on the outer surface as is found in EVs from tumor cells or apoptotic bodies. Can you explain better?

Minor revisions

  1. Please explain all acronyms at their first use
  2. Check the spelling
  3. In the text, reference numbers should be placed in square brackets [ ], and placed before the punctuation.
  4. There are several language problems to correct.
  5. In materials and methods better specify the choice of antibodies for immunofluorescence (as done in the previous work) because for the same cell line you used different antibodies
  6. Insert in all article the centigrade degree symbol
  7. Line 102: 200-250 gm or g?
  8. Replace 1° and 2° with primary and secondary
  9. Line 176: RGM medium
  10. Line 197: it is not clear the concentration of EVs
  11. Line 234: Replace anti-Brna-3a with Anti Brn-3a
  12. Line 237 and 242: please explain what means “in a vesicular staining pattern”?
  13. In line 274 Orientations in C means the cells layer like in A?
  14. In Lines 297-298 and 306-306 EVs are differently represented (concentrated in large vesicles or are large vesicles?)
  15. Insert bar scale in the captions
  16. In Figure 9 indicate the panel with A, B, C etc
  17. Lines 390-392. The paragraph is not clear
  18. Line 405. Maybe it is necessary to also insert the astrocytes

Author Response

Thank you for your helpful review. Responses to  your comments appear below each comment. In the manuscript, any changes made in response to your comments appear in LIGHT BLUE highlighting. 

This reviewer will agree with the publication of this manuscript only after a major revision. There are 3 major points and several minor points to be solved, as follows:

  1. The objectives of this study were not clearly stated. It was the intention of the authors to compare into this paper some results of their current work with other previously published results?< The objectives have now been clearly stated. 
  2. The confocal microscopy figures are difficult to understand because the dimensions are too small and do not allow a clear interpretation (Figures 1,2,3,4, 9,10). Furthermore in Figure 6 the fluorescence decay curve is not visible. The captions are very lacking especially in Figures 1 and 2. The control in PBS is mentioned only in Figure 3 and not in Figures 1 and 2. This reviewer recommends showing only the most significant images of each experiment in the article. <The figures have been revised, making them larger, to enable readers to see the details. Captions have been revised. 
  3. In the discussion is not clearly stated whether EVs from hBMSCs exhibit phosphatidylserine on the outer surface as is found in EVs from tumor cells or apoptotic bodies. Can you explain better? < This point is explained in the Discussion and referenced. 

Minor revisions

  1. Please explain all acronyms at their first use < Done
  2. Check the spelling< Checked first time, rechecked now, and no errors
  3. In the text, reference numbers should be placed in square brackets [ ], and placed before the punctuation.<Done
  4. There are several language problems to correct.<Done
  5. In materials and methods better specify the choice of antibodies for immunofluorescence (as done in the previous work) because for the same cell line you used different antibodies<Antibodies for in vivo are the same as before, and all primary antibodies are shown in a Suppl Table so as not clutter the Methods section and make it harder to read. 
  6. Insert in all article the centigrade degree symbol< Done
  7. Line 102: 200-250 gm or g?<Done
  8. Replace 1° and 2° with primary and secondary<Done
  9. Line 176: RGM medium<Explained what it is
  10. Line 197: it is not clear the concentration of EVs<Done
  11. Line 234: Replace anti-Brna-3a with Anti Brn-3a< Done
  12. Line 237 and 242: please explain what means “in a vesicular staining pattern”?<Explanation provided
  13. In line 274 Orientations in C means the cells layer like in A?<Explanation provided, yes
  14. In Lines 297-298 and 306-306 EVs are differently represented (concentrated in large vesicles or are large vesicles?)<Updated
  15. Insert bar scale in the captions<Done
  16. In Figure 9 indicate the panel with A, B, C etc<I think the numbering is better for this one. 
  17. Lines 390-392. The paragraph is not clear<This has been re-written for clarity
  18. Line 405. Maybe it is necessary to also insert the astrocytes. <Agreed. That has been done. 

Round 2

Reviewer 2 Report

The authors have significantly improved their manuscript and the figures. The reviewer has only few comments:

  • 3, line 2: typo “form” instead of “from”
  • Figure 9 and 10: It looks like the originally square panels of the figures became rectangular. Please correct this.

Author Response

  • 3, line 2: typo “form” instead of “from”
  • Figure 9 and 10: It looks like the originally square panels of the figures became rectangular. Please correct this.  < Thank you. Both typo and figures have been updated. The two figures are now restored to squares. 

Reviewer 3 Report

The present form of the manuscript is recommended for publication after minor revisions.

In the text, reference numbers should be placed in square brackets [ ], and placed before the punctuation.

In the Introduction the objectives  could be better summarized (not listed)

Check the arrowheads in the figures

Author Response

In the text, reference numbers should be placed in square brackets [ ], and placed before the punctuation. < Thank you, rechecked and corrected

In the Introduction the objectives  could be better summarized (not listed)

< Thank you, rephrased. 

Check the arrowheads in the figures

< Thank you, checked for accuracy. Missing notations about arrowheads were added to figure legends.